# Is Greater Social Support from Parents and Friends Related to Higher Physical Activity Levels among Adolescents?

**DOI:** 10.3390/children10040701

**Published:** 2023-04-10

**Authors:** Edina Maria de Camargo, Cristiane Galvão da Costa, Thiago Silva Piola, Eliane Denise Araújo Bacil, José Francisco López-Gil, Wagner de Campos

**Affiliations:** 1Department of Physical Education, Universidade Federal do Paraná (UFPR), Curitiba 81531-980, Paraná, Brazil; 2Navarrabiomed, Hospital Universitario de Navarra (HUN), Universidad Pública de Navarra (UPNA), IdiSNA, 31008 Pamplona, Spain; 3Department of Environmental Health, T.H. Chan School of Public Health, Harvard University, Boston, MA 02138, USA; 4One Health Research Group, Universidad de Las Américas, Quito 170124, Ecuador

**Keywords:** motor activity, social support, youths

## Abstract

Increasing physical activity levels during adolescence have been put on the agenda by several researchers. This study verified the association between social support from parents and friends and different amounts of moderate-to-vigorous physical activity (MVPA) among adolescents in public school. The present study had a cross-sectional design and included a representative sample of 1984 adolescents (aged 15–17). The ASAFA (*Apoio Social para prática de Atividade Física para Adolescentes*) scale and the QAFA (*Questionário de Atividade Física para Adolescentes*) were used to determine social support and physical activity, respectively. For statistical analysis, a conceptual model for structured equations and weighted least squares mean and variance adjusted were applied. Social support from parents increased the odds of engaging in 180 min/week of MVPA by 46.7%, 47.8% for 300 min/week, and 45.5% for 420 min/week. Social support from friends showed similar relations trends: 23.8% for 180 min/week, 23.6% for 300 min/week, and 21.2% for 420 min/week. Social support from parents and friends increased the probability of adolescents reaching the amounts of physical activity investigated. The results indicate that greater social support (from parents and friends) was associated with a higher level of MVPA in Brazilian adolescents.

## 1. Introduction

The literature has shown evidence that supports and reinforces the need to stay active in adolescence, and the aggravations caused by insufficient levels of physical activity have been one way to strengthen its importance [1,2,3,4,5,6]. Eight out of ten adolescents aged 11–17 years do not engage in adequate physical activity to obtain health benefits (i.e., 60 min/day of moderate-to-vigorous physical activity) [1,6]. In Brazil, 84% of adolescents do not meet the physical activity recommendations, with males being more physically active than females [1]. The difference between males and females could be related to unequal leisure and sports options offered to females and males [7,8], little encouragement and lack of company from family and friends [7,8], lack of structures in cities [9,10,11], barriers to the implementation of policies aimed at promoting physical activity for this part of the population [12,13,14,15], and a lack of social support from parents and friends [16,17,18,19]. For these reasons, there is an urgent call for interventions targeting the promotion of adolescent health [20].

The adoption of such behavior happens in a complex manner and can be predicted/influenced by several factors [14,15], including social support from parents and friends [7,8,16,17,18,19]. The scientific literature indicates that adolescents who perceive themselves as receiving support from parents and friends regarding physical activity practice increase their chances of being more physically active [7,8,16,17,18,19]. While some studies have examined the association between social support and physical activity [7,8,16,17,18,19], no previous study has examined the association considering the different levels of physical activity (i.e., 180, 300, 420 min/week). Although they are closely related, there are important differences between the amounts of physical activity engaged in and the differences between males and females (i.e., females may prefer to engage in physical activity with friends or mothers, males may receive greater support from their parents, and fathers can transport and encourage more males). Thus, it can be speculated that more social support is related to a greater probability of physical activity. For those who do not perform any physical activity, doing a little is beneficial. For those who do a little, increasing it will bring additional benefits. For those who meet the recommendations, increasing the amount will bring even more benefits [20,21]. Receiving social support from parents and friends during this process can increase the level of physical activity in this population [7,8,16,17,18,19]. However, the association between the social support received from parents and friends and the different amounts of physical activity is still unclear.

Considering that physical activity-related behavior is an issue that goes beyond a dichotomous phenomenon summarized as “all or nothing”, that is, all support received for physical activity counts, as well as every amount of physical activity matters [20,21]. Analyzing social support from parents and friends and different amounts of physical activity may increase the understanding of the topic and establish future intervention programs to promote physical activity levels among adolescents [7,8,16,17,18].

The objective of this study was to verify the association between the social support received from parents and friends and different amounts of physical activity in adolescents from public schools.

## 2. Materials and Methods

This is a study with a cross-sectional design conducted in 2018, with a representative sample of students aged 15 to 17 years enrolled in high school (Curitiba, Paraná, Brazil). The study was conducted in accordance with the National Health Council’s guidelines for research involving humans (Resolution No. 466/2012), was approved by the Federal University of Paraná’s Research Ethics Committee (CAAE: 98133218.8.0000.0102) and received the students’, guardians’, or parents’ consent via an informed consent form.

For the sample calculation, a prevalence ratio (PR) = 1.49 was considered for the association between social support and the practice of physical activities, and a confidence level of 95% (*α* = 0.05) with a power of 80% (*β* = 0.20) was used. To correct the error related to the sample selection process, an inflation factor (i.e., design effect) of 1.4 was considered to account for variation in cluster size. Furthermore, an addition of 30% for possible losses and refusals was considered, which determined a minimum sample size for the study of 965 adolescents. These calculations were performed using the G*Power calculator version 3.1.9.4 (G*Power, Dusseldorf, Germany).

The sample was selected from the multistage sampling process: 1st stage, all state schools were stratified by region; 2nd stage, a draw of two schools in each region of the city was carried out; and 3rd stage, a simple random selection of one class from each year of high school was carried out, according to the number of students and gender. All students in each class were invited to participate in the study, excluding those who had physical and/or cognitive limitations that limited the practice of physical activity (informed by the school, *n* = 12) and those who were 18 years old (*n* = 125). Students who did not submit the informed consent form (*n* = 104), those who refused to participate in the study or were absent on the day of collection (*n* = 56), and those who answered the questionnaires incorrectly (*n* = 229) were considered sample losses. The analytic sample was 1984 students.

Analyses of the sampling power, performed a posteriori, showed that this sample could identify prevalence ratios (PR) that were statistically significant above 1.28 as risk and below 0.77, considering a power of 80% (β = 20%), a confidence level of 95% (α = 5%), and a prevalence of 34% of students with low social support who did not perform physical activity.

### 2.1. Gender, Age, Body Mass Index, Biological Maturation, Education, Socioeconomic Status

Gender was self-reported (“male”, “female”), and age was calculated from the date of birth (reported by the adolescent) subtracted from the date of data collection and classified as “15 years”, “16 years”, or “17 years”. The body mass index was calculated from the measured data of body mass (kg) and height (m). Body mass was assessed using a portable digital anthropometric scale (W721, Wiso, Curitiba, Brazil) with a resolution of 100 g and a capacity of 150 kg (kg). The students were barefoot, wearing only light clothing and were instructed to maintain their anatomical position, to stand with their backs to the scale and to distribute their body mass equally between both feet. A portable stadiometer (W721, Wiso, Curitiba, Brazil) was used to measure the total height of the participants. The participants were in the anatomical position with their heads positioned in the Frankfurt plane, and at the moment of measurement, they performed inspiratory apnea. The classification for overweight and obese participants was made using the body mass index z score classifications for each gender and age, proposed by the World Health Organization, as follows: underweight ≤ −2 standard deviation (SD), normal weight ≥ −2 SD and < +1 SD, overweight ≥ +1 SD and < +2 SD, and obesity ≥ +2 SD [22]. Similarly, participants were classified as “not excess weight” (underweight or normal weight) and “excess weight” (overweight or obesity) [22].

Biological maturation was determined by sexual maturation using the Tanner stages [23]. The Tanner stages, through the method of self-assessment of pubic hair on males and females, determine the maturational stages on a scale of 1 to 5, where stage 1 is when the subject is at the level considered prepubertal, intermediate stages 2, 3, and 4 when the subject is during the maturational process (pubertal), and stage 5 when the maturational process is complete (postpubertal). This method was determined by comparative self-assessment with illustrative boards on the appearance of pubic hair for both females and males. Sexual maturation was classified as stage 1 (prepubertal), 2 (pubertal), and 3 (postpubertal); however, after collecting and analyzing the data, only students in the pubertal and postpubertal stages were detected [23].

Parental education and socioeconomic status were assessed by the Brazilian Association of Research Companies [24]. Parental education was classified as elementary school, high school, and university (the schooling of the father, mother, and head of the family was asked, and the schooling of the head of the family was used to calculate the socioeconomic status). Socioeconomic status (SES) was classified into three categories: “low” (class C + D + E), “intermediate” (class B1 +B2), and “high” (class A1 + A2) [24].

### 2.2. Social Support from Parents or Guardians and Friends

Social support (SS) from parents and friends for physical activity was determined by the 10-item ASAFA (*Apoio Social para prática de Atividade Física para Adolescentes*) scale, which showed satisfactory internal consistency (parents: α ≥ 0.77 and CFI ≥ 0.83; friends: α ≥ 0.87 and CFI ≥ 0.91) [25]. Participants indicated the frequency (never = 1, rarely = 2, often = 3, always = 4) when their parents and friends offered them some type of social support for physical activity (to encourage, to practice, to transport, to attend, to comment, to invite) during a typical or normal week [25]. For analysis purposes, the response options were grouped and classified as “receives social support” (rarely, often, always) and “does not receive social support” (never), making it dichotomous.

### 2.3. Adolescents’ Moderate-to-Vigorous Physical Activity (MVPA)

The level of physical activity was measured by the QAFA (*Questionário de Atividade Física para Adolescentes*) [26], developed in a checklist format by Sallis et al. [27]. In Brazilian adolescents, the questionnaire showed good reproducibility (intraclass correlation coefficient = 0.88; 95% CI = 0.84–0.91) and concurrent validity in comparison with 24 h recall (*r* = 0.62; *p* < 0.001) [26]. The questionnaire is composed of a list of 24 physical activities. For analysis, the amount of MVPA performed by adolescents in a usual week was classified as 180 min, 300 min, and 420 min/week. The amount of physical activity was used to construct the dependent variables of whether the students complied with 180 min, 300 min, or 420 min of MVPA. Since the QAFA does not assess the intensity of physical activity, the adolescents were also asked about the intensity of the physical activities they performed.

### 2.4. Data Analysis

Descriptive statistics were performed to verify the sample characteristics and social support (SPSS software version 23). Subsequently, with R software version 3.6.1, confirmatory factor analysis was performed to analyze the one-dimensionality of the constructs included in the analysis. Next, the validity of the conceptual model was tested by calculating the covariances between the constructs. Finally, the measurement model was tested through structural equation modeling analysis. Due to the use of dichotomous variables, the weighted least squares mean and variance adjusted (WLSMV) estimator was used, which provided the calculation for categorical correlations. The model was considered valid when it met the following criteria [28]: (a) value of the chi-square (*χ*^2^) adjustment statistic with *p* > 0.05; (b) root mean square error of approximation (RMSEA) ≤ 0.08; and (c) Tucker–Lewis index (TLI) and comparative fit index (CFI) greater than or equal to 0.90. Both for the creation of the latent variables and for the estimation of the impacts on the variable of interest (engaging in MVPA: 180 min, 300 min, and 420 min), logistic models with binary response variables were used after CFA (confirmatory factor analysis) to test the suitability of the data analysis. The use of logistic models allowed for the interpretation of the results in terms of probability, indicating how the independent variables explain the probability of engaging in MVPA. A *p* < 0.05 was adopted for all the analyses.

## 3. Results

The sample was composed of 1984 students (female *n* = 1109; 55.9%). The average age was 16.01 years (SD = 0.80), and the highest frequency of adolescents reported being in the postpubertal stage with respect to sexual maturation (95.2% of the sample, 96.7% males and 94% females). The average body weight was 59.65 kg (SD = 11.09), and the average height was 1.68 m (SD = 0.08). Regarding the classification of overweight and obesity, 12.3% of the sample was classified as overweight, with 15.0% males and 10.2% females. Most adolescents had parents who had completed high school education and belonged to the intermediate socioeconomic status group (Table 1).

Regarding the practice of moderate-to-vigorous physical activity MVPA, 21.6% (*n* = 429) of males and 24.4% (*n* = 483) of females reported doing 420 min/week; 25.0% (*n* = 495) of males and 28.9% (*n* = 572) of females reported doing 300 min/week; and 28.7% (*n* = 568) of males and 33.2% (*n* = 659) of females reported doing 180 min/week. Regarding social support (SS), 43.5% (863) of adolescents mentioned receiving social support from their parents for physical activity (19.9% or 394 males; 23.6% or 469 females). In addition, 45.6% (905) mentioned receiving (some) social support from friends to practice physical activity (21.4% or 424 males; 24.2% or 481 females). Other information about social support is shown in Table 2.

Table 2 presents data on the social support received from parents and friends for the practice of physical activity according to the adolescent’s gender.

Social support from parents/guardians leads to an increase in the probability of engaging in 180 min of MVPA per week by approximately 46.7% (*p* < 0.001). Males are on average 2.2% more likely to perform 180 min of MVPA per week, all else being constant. The model showed good measures for quality of fit. The TLI and CFI values above 0.9 indicate that the model was able to improve the explanatory power of MVPA variability by more than 90%. RMSEA and SRMR values less than 0.08 indicate that the model was a good fit in the sense that the residuals left by the model were low, indicating good predictive ability (Figure 1).

Social support from parents/guardians leads to an increase in the probability of performing 300 min of MVPA per week by approximately 47.8% (*p* < 0.001). Gender was not significant in explaining participating (or not) in MVPA for 300 min per week. The model showed good measures for quality of fit. The TLI and CFI values above 0.9 indicate that the model was able to improve the explanatory power of MVPA variability by more than 90%. RMSEA and SRMR values less than 0.08 indicate that the model was a good fit in the sense that the residuals left by the model were low, indicating good predictive ability (Figure 1).

Social support from parents/guardians leads to an increase in the probability of performing 420 min of MVPA per week by approximately 45.5% (*p* < 0.001). Males are on average 2.6% more likely to perform 420 min of MVPA per week, all else being constant (*p* < 0.001). The model showed good measures for quality of fit. The TLI and CFI values above 0.9 indicate that the model was able to improve the explanatory power of MVPA variability by more than 90%. RMSEA and SRMR values less than 0.08 indicate that the model was a good fit in the sense that the residuals left by the model were low, indicating good predictive ability (Figure 1).

Social support from friends leads to an increase in the probability of engaging in 180 min of MVPA per week by approximately 23.8% (*p* < 0.001). Males are on average 2.8% more likely to perform 180 min of MVPA per week, all else being constant (significant at 1%). The model showed good measures for quality of fit. The TLI and CFI values above 0.9 indicate that the model was able to improve the explanatory power of MVPA variability by more than 90%. RMSEA and SRMR values less than 0.08 indicate that the model was well fitted in the sense that the residuals left by the model were low, which shows good predictive ability (Figure 2).

Social support from friends leads to an increase in the probability of engaging in 300 min of MVPA per week by approximately 23.6% (*p* < 0.001). Males are on average 2.1% more likely to perform 300 min of MVPA per week, all else being constant (*p* < 0.001). The model showed good measures for quality of fit. The TLI and CFI values above 0.9 indicate that the model was able to improve the explanatory power of MVPA variability by more than 90%. RMSEA and SRMR values less than 0.08 indicate that the model was well fitted in the sense that the residuals left by the model were low, which shows good predictive ability (Figure 2).

Social support from friends leads to an increase in the probability of engaging in 420 min of MVPA per week by approximately 21.2% (*p* < 0.001). Males are on average 3.2% more likely to perform 420 min of MVPA per week, all else being constant (*p* < 0.05). The model showed good measures for quality of fit. The TLI and CFI values above 0.9 indicate that the model was able to improve the explanatory power of MVPA variability by more than 90%. RMSEA and SRMR values less than 0.08 indicate that the model was well fitted in the sense that the residuals left by the model were low, which shows good predictive ability (Figure 2).

Although our estimated models did not reach a CFI ≥ 0.90 and a TLI ≥ 0.90, we obtained acceptable values for chi-square and the RMSEA. The model fit index values of CFI and TLI are commonly used above 0.90. Table 3 shows the relationship between social support from parents and friends and different amounts of physical activity (180, 300, and 420 min/week). Gender was included as a control variable.

## 4. Discussion

Our findings strengthen the premise that different social contexts foster different types of behaviors [14,15,29,30]. Receiving support from parents and friends for physical activity practice, regardless of the amount of physical activity performed by adolescents in a usual week, increases the chances of practicing physical activity. The probability did not increase linearly regarding any of the sources (parents and friends).

Support from parents has been related to, more than the support received from friends, the probability of adolescents engaging in physical activity, regardless of the amount performed, 180, 300 or 420 min/week, which strengthens the results found in the literature [7,8,17,18]. Otherwise, studies have shown that during childhood, children more frequently perceive the social support received from parents, and in adolescence, they more frequently perceive the support received from friends [16]. It is noteworthy that, regardless of adolescence being a period of greater family independence, their social support seems to play an important role in increasing the level of physical activity, which highlights the need for family support, especially regarding transportation, positive feedback for physical activity, and logistic support for transport [7,8,16,17,18,19,31,32]. The studies suggest that this family support may vary according to the region of the country [31,32], which may partially explain the difference found in the data obtained among adolescents from Curitiba (Paraná, Brazil).

Although adolescence experiences vary according to context [19], the family promotes changes in behavioral patterns, and adolescents who perceive support for their physical activity practice end up adopting healthy habits and becoming physically active [32,33,34,35,36]. The family, especially parents, teaches skills and holds beliefs that help adolescents shape important health attitudes and behaviors, most specifically concerning physical activity and reducing sedentary behavior [32,33,34,35,36]. In a systematic review of the reviews [36], in which the included studies were rated according to very high quality and relevance, the evidence revealed the key role played by parents in promoting physical activity during childhood and adolescence.

As a practical application, our results seem to highlight the need for parents to encourage adolescents to be physically active in their leisure time [7,8,16]. Suggesting changes in the adolescent’s daily routine aimed at maximizing energy expenditure may be a good alternative to help adolescents realize opportunities to be active during their daily routine, for example, going out at the end of the day on a skateboard, bicycle, or foot; exercising in public leisure spaces; or even visiting public spaces during outings with family and friends [7,16,19,31,32]. The incentive for the practice of MVPA, even with small insertions in the daily routine, may contribute to increasing the amount of physical activity in a usual week [7,8,16].

Social support from friends has been shown to maximize physical activity levels in adolescents [7,16,31,32], which strengthens the results found in the literature [7,8,17,18,31,32,33,34,35,36]. In a 12-month study, Lawler et al. [33] highlighted the need to consider both sources of social support (parents and friends) to change adolescent behavior, claiming that different socializing agents transmit different influences. Future studies should investigate whether teenagers realize social support differently from fathers, mothers, grandmothers, grandfathers, teachers, coaches, and friends [32,35]. In the present study, gender was included as a control variable and did not explain adherence to physical activity. In all the investigated physical activity amounts, males who perceived social support from parents and friends were more likely to perform physical activity than females. Interventions targeting the promotion of females’ health are needed, and social support from parents and friends can be an important predictor of females’ physical activity [7,8].

As a practical application, our results seem to highlight the need for group discussions, conversations, e-mail messages and apps, and texts on healthy behavior, which can be efficient strategies to disseminate the practice. Verbal persuasion activities that prompt adolescents to express their beliefs in their ability to adopt the behavior may support a better understanding of actual outcome expectations. Another suggested strategy is to visualize people or groups exercising in public places, which can increase the motivation to perform the activity [9,10]. These techniques can be introduced in interventions that aim to increase physical activity practice in different contexts, such as physical education classes at school, MVPA, active commuting to school, and active leisure time at school [19,37]. School is the place where adolescents spend most of their time away from home and in the company of their friends [37]. Social support plays an important role in the relationship with physical activity since friends share ideas, habits, and beliefs, which can increase the amount of physical activity when friends are also physically active [19,37].

Some limitations should be considered for a better understanding of the results: (1) Reverse causality does not allow for establishing a cause-and-effect relationship or the direction of relationships. (2) The study was developed in a single Brazilian city that presents typical characteristics of well-developed urban centers, which does not allow for the extrapolation of the findings to rural centers. (3) The sample consisted only of students from public schools, which makes it impossible to extrapolate the results to higher classes. However, the representative sample and statistical analyses ensure an interpretation of the data for large populations. (4) The use of self-reported measures could be influenced by recall bias. To minimize this bias, the researchers were trained to assist students in answering the questionnaires. Because this was a large study with a representative sample, the use of questionnaires was a feasible option.

Our findings are important for the design of future interventions aimed at changing behavior in the amount of physical activity involving adolescents. As a public health message, the implications of this study highlight a set of actions aimed at disseminating and maximizing the practice of physical activity among adolescents. As examples of effective strategies to encourage the practice of physical activity, the following are proposed: (a) education by example aimed at incorporating attitudes and behaviors that facilitate physical activity and reduce sedentary behavior; (b) guidance to perceive opportunities to be physically active in their neighborhood with their friends through awareness of new public places; and (c) participation in social groups involving discussion of the topic through conversations, messages, informative texts, and the use of social media for interaction with people of the same age on topics relevant to physical activity, exercise, and sports. As our results suggest, some advances could be achieved if interventions with adolescents consider the importance of social support from parents and friends to practice physical activity, such as encouraging, practicing together, providing transportation, attending, commenting on physical activity, and inviting adolescents to perform MVPA.

## 5. Conclusions

The results indicate that greater social support (from parents and friends) was associated with a higher level of MVPA in Brazilian adolescents. Parents and friends should encourage, practice together, provide transportation, attend, comment on physical activity, and invite adolescents to perform MVPA. This can contribute to a change in the behavior of this population. With the support of parents and friends, adolescents who do not perform any physical activity can start doing a little. Those who do a little can increase their activity and gain additional benefits. For those who meet the minimum health-related recommendations, increasing the amount of physical activity will bring even more benefits.

## Figures and Tables

**Figure 1 children-10-00701-f001:**
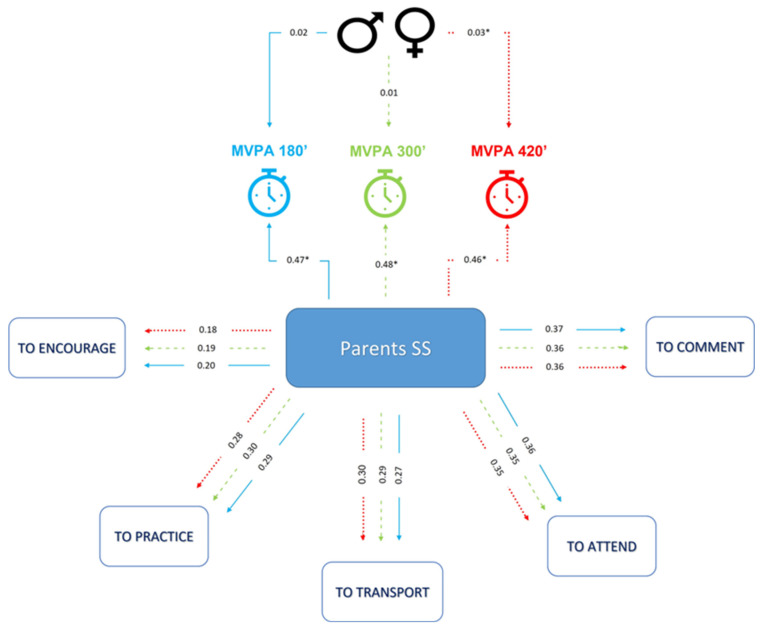
Association of social support received from parents with different levels of moderate-to-vigorous physical activity performed by adolescents. MVPA: moderate-to-vigorous physical activity; SS: social support. TLI and CFI > 0.9. RMSEA and SRMR < 0.08. * *p* value < 0.001.

**Figure 2 children-10-00701-f002:**
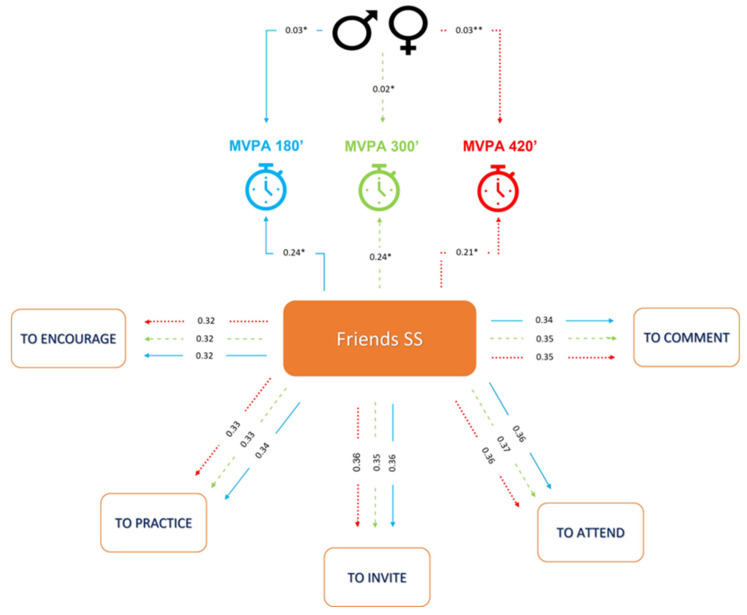
Association of social support received from friends with different levels of moderate-to-vigorous physical activity performed by adolescents. MVPA: moderate-to-vigorous physical activity; SS: social support. TLI and CFI > 0.9. RMSEA and SRMR < 0.08. * *p* value < 0.001; ** *p* value < 0.05.

**Table 1 children-10-00701-t001:** Description of the sociodemographic characteristics of the adolescents from public schools (*N* = 1984).

Variables		Male (*n* = 875; 44.1%)		Female (*n* = 1109; 55.9%)			Total	
		*n*	%	*n*	%	*p* *	*n*	%
Age	15 years	261	41.8	363	58.2	0.124	624	31.5
	16 years	317	44.3	399	55.7		716	36.1
	17 years	297	46.1	347	53.9		644	32.4
BMI ^¥^	No excess weight	744	85.0	996	89.8	0.001	1740	87.7
	Excess weight	131	15.0	113	10.2		244	12.3
Father’s Education	Up to Elementary School	238	38.1	387	61.9	0.003	625	31.5
	High School	405	47.2	453	52.8		858	43.2
	University	232	46.3	269	53.7		501	25.3
Mother’s Education	Up to Elementary School	245	39.0	384	61.0	0.006	629	31.7
	High School	398	46.3	461	53.7		859	43.3
	University	232	46.8	264	53.2		496	25.0
SES	Low	140	40.0	210	60.0	0.005	350	17.6
	Intermediate	538	43.4	702	56.6		1240	62.5
	High	197	50.0	197	50.0		394	19.9

SES: social economic status. * *p* value of the chi-square test; **^¥^** According to the World Health Organization criteria [22].

**Table 2 children-10-00701-t002:** Social support (parents and friends) for the practice of physical activity according to gender (*N* = 1984).

Social Support		Male (*n* = 875; 44.1%)		Female (*n* = 1109; 55.9%)			Total		
		*n*	%	*n*	%	*p* *	*n*	%	%
Parents: To Encourage	Never	170	19.4	240	21.6	0.285	410	20.7	100
	Sometimes	473	54.1	604	54.3		1077	54.3	
	Always	232	26.5	265	23.9		497	25.0	
Parents: To Practice	Never	322	36.8	455	41.0	0.086	777	39.2	100
	Sometimes	434	49.6	530	47.8		964	48.6	
	Always	119	13.6	124	11.2		243	12.2	
Parents: To Transport	Never	425	48.6	541	48.8	0.443	966	48.7	100
	Sometimes	262	29.9	353	31.8		615	31.0	
	Always	188	21.5	215	19.4		403	20.3	
Parents: To Attend	Never	370	42.3	500	45.1	0.382	870	43.9	100
	Sometimes	365	41.7	450	40.6		815	41.1	
	Always	140	07.1	159	08.0		299	15.0	
Parents: To Comment	Never	324	37.0	449	40.5	0.059	773	39.0	100
	Sometimes	341	39.0	375	33.8		716	36.1	
	Always	210	24.0	285	25.7		495	24.9	
Friends: To Encourage	Never	324	37.0	445	40.1	0.370	769	38.8	100
	Sometimes	355	40.6	426	38.4		781	39.4	
	Always	196	22.4	238	21.5		434	21.8	
Friends: To Practice	Never	229	26.2	348	31.4	0.019 *	577	29.1	100
	Sometimes	370	42.3	460	41.5		830	41.8	
	Always	276	31.5	301	7.1		577	29.1	
Friends: To Invite	Never	243	27.8	376	33.9	0.014 *	619	31.2	100
	Sometimes	381	43.5	440	39.7		821	41.4	
	Always	251	28.7	293	26.4		544	27.4	
Friends: To Attend	Never	421	48.1	520	46.9	0.748	941	47.4	100
	Sometimes	309	35.3	410	37.0		719	36.2	
	Always	145	16.6	179	16.1		324	16.4	
Friends: To Comment	Never	425	48.6	566	51.0		991	49.9	100
	Sometimes	308	35.2	363	32.7		671	33.9	
	Always	142	16.2	180	16.2		322	16.2	

Note: “Sometimes” groups include “rarely” and “often”. * *p* value of the chi-square test.

**Table 3 children-10-00701-t003:** Association of social support from parents and friends with the moderate-to-vigorous physical activity of adolescents (*N* = 1984).

Physical Activity Duration	Social Support	Direct Effect	*p*
180 min/week	Parents	0.1467	<0.001
	Gender	0.022	0.080
300 min/week	Parents	0.478	<0.001
	Gender	0.014	0.140
420 min/week	Parents	0.455	<0.001
	Gender	0.026	<0.001
180 min/week	Friends	0.238	<0.001
	Gender	0.028	<0.001
300 min/week	Friends	0.236	<0.001
	Gender	0.021	<0.001
420 min/week	Friends	0.212	<0.001
	Gender	0.032	0.020

## Data Availability

The data presented in this study are available on request from the corresponding author.

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
