# Peer review of "Is Greater Social Support from Parents and Friends Related to Higher Physical Activity Levels among Adolescents?"

_children, 2023, doi:10.3390/children10040701_

Round 1

Reviewer 1 Report

The authors did an excellent work to investigate the relationship between social support and PA among adolescents. A related factor identified by regression analysis would be more plausible.

Author Response

The authors did an excellent work to investigate the relationship between social support and PA among adolescents. A related factor identified by regression analysis would be more plausible.

Answer= Thank you so much for your time and feedback. We appreciate the reviewer’s suggestion. However, we consider it more interesting to report our results in that way (i.e., SEM analysis).

Attachment Letter to Reviewers*

Reviewer 2 Report

Was the research conducted in a secret location? It looks like it! The questionnaire does not have an eloquent description... The results of the research, even if they are the result of fashionable statistics, are more poetic.

Author Response

Was the research conducted in a secret location? It looks like it! The questionnaire does not have an eloquent description... The results of the research, even if they are the result of fashionable statistics, are more poetic.

Answer= Thank you for your comment. The research was conducted in Curitiba (Brazil). It has been added. In addition, the questionnaire has been further explained as follows:

Page 3= “Social support (SS) from parents and friends for physical activity was measured by a 10-item scale (ASAFA Scale), which shows satisfactory internal consistency (parents: α ≥ 0.77 and CFI ≥ 0.83; friends: α ≥ 0.87 and CFI ≥ 0.91) [25]”.

Page 4: “The level of physical activity of adolescents was measured by the Questionnaire of Physical Activity for Adolescents (QAFA) [26], which was first developed in a checklist format by Sallis et al. [27] for US adolescents, with later translation and adaptation for Brazilian adolescents by Farias-Júnior and collaborators [26]. The questionnaire is composed of a list of 24 moderate to vigorous physical activities, with the possibility of the adolescent adding activities beyond those listed. In Brazilian adolescents, the questionnaire showed good reproducibility (intraclass correlation coefficient = 0.88; 95% CI = 0.84-0.91) and concurrent validity in comparison with 24-hour recall (r = 0.62; p < 0.001) [26].

Attachment Letter to Reviewers*

Reviewer 3 Report

The title is in the form of a question. It should be modified and written based on the objective of the study.

Eliminate the image that appears before the abstract.

The introduction is extremely brief; more research should be done on academic references that deal with physical activity levels and their possible relationship with social support.

Are there specific objectives?

The methodological procedure should be more detailed.

The methodology should present hypotheses and justify them.  

There is a lack of more graphs and tables that achieve understanding of the research.

Conclusions are very brief There is a lack of information regarding the research objectives.

Author Response

The title is in the form of a question. It should be modified and written based on the objective of the study. Eliminate the image that appears before the abstract. The introduction is extremely brief; more research should be done on academic references that deal with physical activity levels and their possible relationship with social support. Are there specific objectives? The methodological procedure should be more detailed. The methodology should present hypotheses and justify them. There is a lack of more graphs and tables that achieve understanding of the research. Conclusions are very brief There is a lack of information regarding the research objectives.

Answer = We appreciate your suggestion. We have considered that the title is in an adequate way. This journal accepts articles with the title in that way. In addition, we have removed the image that appears before the abstract.

The introduction section has been increased, and we have included more references related to social support. On the other hand, we have no specific goals in this study, only a general goal. Moreover, the hypothesis and justification were included. Changes Included:

Page 1=“The difference between sexes may occur because of the unequal leisure and sports options offered to girls and boys [7,8], little encouragement and lack of company from family and friends [7-8], lack of structures in cities [9-11] or even barriers to the implementation of policies aimed at promoting physical activity for this part of the population [12-15], lack of social support from parents and friends [16-19]. For these reasons, there is an urgent call for interventions targeting the promotion of adolescent health [20].

Page 2= The scientific literature indicates that adolescents who perceive support from parents and friends for physical activity practice increase their chances of being more physically active [7,8,16-19]. While some studies have examined the association between social support and physical activity [7,8,16-19], no previous study has examined the association between considering the different levels of physical activity (i.e., 180, 300, 420 min/week). Although they are closely related, there are important differences between amounts of physical activity and differences between boys and girls (i.e., girls may prefer to do physical activity with the company of friends or mothers; boys may receive greater support from their parents; fathers can transport and encourage more boys). Thus, it can be speculated that a greater amount of social support is related to a greater probability of physical activity”.

Page 2= However, the association between social support received from parents and friends and different amounts of physical activity is still scarce.

Considering that physical activity-related behavior is an issue that goes beyond a dichotomous phenomenon summarized as "all or nothing", that is, every support received for physical activity counts, as well as every amount of physical activity matters [20,21]. Analyzing social support from parents and friends with different amounts of physical activity may increase understanding of the topic and establish future intervention programs to promote physical activity levels among adolescents [7,8,16-18].

Added references=

Garcia-Hermoso A, López-Gil JF, Ramírez-Vélez R, et al. Adherence to aerobic and muscle-strengthening activities guidelines: a systematic review and meta-analysis of 3.3 million participants across 32 countries. British Journal of Sports Medicine 2023;57:225-229.

GBD 2019 Mental Disorders Collaborators. Global, regional, and national burden of 12 mental disorders in 204 countries and territories, 1990e2019: a
systematic analysis for the global burden of disease study 2019. Lancet Psy- chiatr 2022;9(2):137e50.

GBD 2019 Diseases and Injuries Collaborators. Global burden of 369 diseases and injuries in 204 countries and territories, 1990-2019: a systematic analysis for the Global Burden of Disease Study 2019. Lancet. 2020 Oct 17;396(10258):1204-1222. doi: 10.1016/S0140-6736(20)30925-9. Erratum in: Lancet. 2020 Nov 14;396(10262):1562. PMID: 33069326; PMCID: PMC7567026.

Mendonça G, Cheng LA, Mélo EN, de Farias Júnior JC. Physical activity and social support in adolescents: a systematic review. Health Educ Res. 2014 Oct;29(5):822-39. doi: 10.1093/her/cyu017. Epub 2014 May 8. PMID: 24812148.

Cheng LA, Mendonça G, Farias Júnior JC. Physical activity in adolescents: analysis of the social influence of parents and friends. J Pediatr (Rio J). 2014 Jan-Feb;90(1):35-41. doi: 10.1016/j.jped.2013.05.006. Epub 2013 Oct 22. PMID: 24156835.

Su DLY, Tang TCW, Chung JSK, Lee ASY, Capio CM, Chan DKC. Parental Influence on Child and Adolescent Physical Activity Level: A Meta-Analysis. Int J Environ Res Public Health. 2022 Dec 15;19(24):16861. doi: 10.3390/ijerph192416861. PMID: 36554746; PMCID: PMC9778652.

Camargo EM de, Santos MPM, Ribeiro AGP, Mota J, Campos W de. Interação dos fatores sociodemográficos na associação entre fatores psicossociais e transporte ativo para a escola. Cad Saúde Pública [Internet]. 2020;36(Cad. Saúde Pública, 2020 36(5)):e00102719. Available from: https://doi.org/10.1590/0102-311X00102719

The methods were better described:

Page 3= “Social support (SS) from parents and friends for physical activity was measured by a 10-item scale (ASAFA Scale), which shows satisfactory internal consistency (parents: α ≥ 0.77 and CFI ≥ 0.83; friends: α ≥ 0.87 and CFI ≥ 0.91) [25]”.

Page 4: “The level of physical activity of adolescents was measured by the Questionnaire of Physical Activity for Adolescents (QAFA) [26], which was first developed in a checklist format by Sallis et al. [27] for US adolescents, with later translation and adaptation for Brazilian adolescents by Farias-Júnior and collaborators [26]. The questionnaire is composed of a list of 24 moderate to vigorous physical activities, with the possibility of the adolescent adding activities beyond those listed. In Brazilian adolescents, the questionnaire showed good reproducibility (intraclass correlation coefficient = 0.88; 95% CI = 0.84-0.91) and concurrent validity in comparison with 24-hour recall (r = 0.62; p < 0.001) [26].

Page 4: “Due to the use of dichotomous variables, the Weighted Least Square means and variance adjusted (WLSMV) estimator was used, which provides the calculation for categorical correlations. The model was considered valid when it met the following criteria [28]: value of the chi-square (X2) adjustment statistic with p > 0.05; Root mean error of approximation (RMSEA) ≤ 0.05. At the literature-recommended measures of Comparative Fit index (CFI) ≥ 0.90 and Tucker‒Lewis index (TLI)) ≥ 0.90, the estimated models did not reach such high values, but acceptable values for chi-square and RMSEA. The model fit index values of CFI and TLI are commonly used above 0.90. The mean square error of approximation (RMSEA) must be less than 0.08, which indicates an acceptable fit of the model. Values less than 0.05 are considered very good. The comparative fit index (CFI) must be greater than 0.90, which indicates a good model fit [28].

28-Kline, R. B. (2016). Principles and practice of structural equation modeling (4th ed.) New York, NY The Guilford Press.

A table with information on social support was included (table 2).

More information about the research objectives, practical applications and conclusion was better described:

Page 09: “As a practical application, our results seem to highlight the need: parents should encourage adolescents to be physically active in their leisure time [7,8,16]. Suggesting changes in the adolescent's daily routine aimed at maximizing energy expenditure may be a good alternative to help adolescents realize opportunities to be active during their daily routine, for example, going for a walk at the end of the day on a skateboard, bicycle, on foot; exercising in public leisure spaces; or even visiting public spaces through outings with family and friends [7,16,19,31,32]. The incentive for the practice of physical activity in leisure time, even with small insertions in the daily routine, may contribute to increasing the amount of physical activity in a usual week [7,8,16]”.

Page 09: “Social support from friends has been shown to maximize physical activity levels in adolescents [7,16,31,32], which strengthens the results found in the literature [7,8,17,18, 31-36]. In a 12-month study, Lawler et al. [33] highlighted the need to consider both sources of social support (parents and friends) to change adolescent behavior, aiming that different socializing agents transmit different influences. Future studies should investigate whether teenagers differently realize the social support received from fathers, mothers, grandmothers, grandfathers, teachers, coaches, and friends [32,35]. In the present study, gender was included as a control variable and did not explain adherence to physical activity in all the investigated physical activity amounts. Boys who perceived social support from parents and friends were more likely to perform physical activity than girls. Interventions targeting the promotion of girls' health are needed, and social support from parents and friends can be an important predictor of girls' physical activity [7,8].”

Page 10: “The results indicate that social support (parental and friends) increased the likelihood of performing leisure-time physical activity in Brazilians adolescents, regardless of the amount of physical activity analyzed. Parents and friends should encourage, practice together, provide transportation, assist, comment on physical activity, and invite adolescents to perform physical activity in their leisure time. This can contribute to a change in the behavior of this population. With the support of parents and friends, adolescents who do not perform any physical activity can start doing a little. Those who do a little can increase and have additional benefits. For those who meet the minimum health-related recommendations, increasing the amount will bring even more benefits.”

 *Attachment Letter to Reviewers

Reviewer 4 Report

Dear Author,

Overall, the Introduction section provides a good overview of the problem and the research question. However, there are a few areas where the manuscript could be improved:

  1. The introduction could benefit from a more explicit statement of the research question or hypothesis. Currently, it is implied but not clearly stated.

  2. The literature review could be more comprehensive. While the authors provide a good overview of the existing literature on physical activity and social support, they could expand on this and provide more context for the study. For example, they could discuss the impact of physical activity on health outcomes, the importance of promoting physical activity in adolescence, and the challenges associated with increasing physical activity levels in this population.

  3. The authors could provide more information on the limitations of previous research and how this study will address these limitations.

  4. The Methods section is well-written and provides all the necessary details for the study. However, the authors could benefit from providing more information on the instruments used to measure physical activity and social support. They could also discuss the validity and reliability of these instruments.

  5. The authors could provide a more detailed description of the statistical methods used in the analysis. For example, it is not clear how the authors determined which categories showed the greatest difference in proportions between the sexes when using Pearson's chi-square test. A more detailed explanation of the post-hoc tests used would help readers understand the results better.

  6. The authors could provide more information on the confirmatory factor analysis (CFA) and structural equation modeling (SEM) procedures. For example, it is not clear how the authors determined the number of factors to extract or the justification for using a particular SEM model. A more detailed description of the CFA and SEM procedures would help readers understand the validity of the model.

  7. The authors could provide more information on the Weighted Least Square means and variance adjusted (WLSMV) estimator used in the analysis. For example, it is not clear how this estimator handles categorical variables or why it was chosen for this analysis.

  8. The authors could provide more details on the complex sampling plan used to correct for clustering effects. For example, it is not clear how the authors determined the codes used for administrative regions and schools.

  9. The authors could provide more detail about the specific findings of their study, rather than simply stating that the results are in line with those published in the literature. For example, they could describe the magnitude of the effects observed and any notable patterns in the data.

  10. The authors could more explicitly address the limitations of their study and the implications of these limitations for the interpretation of their results. For example, they could discuss how the cross-sectional design of the study limits the ability to draw causal conclusions and the potential impact of selection bias due to the use of a convenience sample.

  11. The authors could provide more context for their findings by discussing how they relate to previous research in the field. For example, they could compare their results to those of other studies that have examined the role of social support in promoting physical activity among adolescents, and they could discuss any inconsistencies or discrepancies between their findings and those of previous research.

  12. The authors could more explicitly state the practical implications of their findings for public health policy and intervention development. For example, they could discuss how their findings could inform the design of programs aimed at promoting physical activity among adolescents, and they could suggest specific strategies that could be employed to increase social support for physical activity among parents and friends.

  13.  

Overall, the manuscript appears to be well-written and presents important findings on the association between social support and physical activity among adolescents. The study design, including a representative sample of 1984 adolescents and the use of validated scales for social support and physical activity, adds credibility to the results.

One potential improvement for the manuscript would be to provide more detailed information on the conceptual model for structured equations that was used in the statistical analysis. This would help readers better understand how the authors arrived at their conclusions.

Additionally, while the study found that social support from parents and friends increased the likelihood of adolescents engaging in different amounts of leisure-time physical activity, the manuscript could benefit from a discussion on the implications of these findings. For example, what strategies could parents and friends use to provide incentives and support for physical activity? How can schools and public health professionals promote social support for physical activity among adolescents?

Overall, this manuscript is a valuable contribution to the field of adolescent physical activity and social support, and the authors should be commended for their work.

Author Response

Dear Author, Overall, the Introduction section provides a good overview of the problem and the research question. However, there are a few areas where the manuscript could be improved:

  1. The introduction could benefit from a more explicit statement of the research question or hypothesis. Currently, it is implied but not clearly stated.
  2. The literature review could be more comprehensive. While the authors provide a good overview of the literature on physical activity and social support, they could expand on this and provide more context for the study. For example, they could discuss the impact of physical activity on health outcomes, the importance of promoting physical activity in adolescence, and the challenges associated with increasing physical activity levels in this population.
  3. The authors could provide more information on the limitations of previous research and how this study will address these limitations.

Answer = We appreciate your suggestion. The introduction section has been increased, and we have included more references related to social support. Moreover, the hypothesis and justification were included. Changes Included:

Page 1=“The difference between sexes may occur because of the unequal leisure and sports options offered to girls and boys [7,8], little encouragement and lack of company from family and friends [7-8], lack of structures in cities [9-11] or even barriers to the implementation of policies aimed at promoting physical activity for this part of the population [12-15], lack of social support from parents and friends [16-19]. For these reasons, there is an urgent call for interventions targeting the promotion of adolescent health [20].

Page 2= The scientific literature indicates that adolescents who perceive support from parents and friends for physical activity practice increase their chances of being more physically active [7,8,16-19]. While some studies have examined the association between social support and physical activity [7,8,16-19], no previous study has examined the association between considering the different levels of physical activity (i.e., 180, 300, 420 min/week). Although they are closely related, there are important differences between amounts of physical activity and differences between boys and girls (i.e., girls may prefer to do physical activity with the company of friends or mothers; boys may receive greater support from their parents; fathers can transport and encourage more boys). Thus, it can be speculated that a greater amount of social support is related to a greater probability of physical activity”.

Page 2= However, the association between social support received from parents and friends and different amounts of physical activity is still scarce.

Considering that physical activity-related behavior is an issue that goes beyond a dichotomous phenomenon summarized as "all or nothing", that is, every support received for physical activity counts, as well as every amount of physical activity matters [20,21]. Analyzing social support from parents and friends with different amounts of physical activity may increase understanding of the topic and establish future intervention programs to promote physical activity levels among adolescents [7,8,16-18].

Added references=

Garcia-Hermoso A, López-Gil JF, Ramírez-Vélez R, et al. Adherence to aerobic and muscle-strengthening activities guidelines: a systematic review and meta-analysis of 3.3 million participants across 32 countries. British Journal of Sports Medicine 2023;57:225-229.

GBD 2019 Mental Disorders Collaborators. Global, regional, and national burden of 12 mental disorders in 204 countries and territories, 1990e2019: a
systematic analysis for the global burden of disease study 2019. Lancet Psy- chiatr 2022;9(2):137e50.

GBD 2019 Diseases and Injuries Collaborators. Global burden of 369 diseases and injuries in 204 countries and territories, 1990-2019: a systematic analysis for the Global Burden of Disease Study 2019. Lancet. 2020 Oct 17;396(10258):1204-1222. doi: 10.1016/S0140-6736(20)30925-9. Erratum in: Lancet. 2020 Nov 14;396(10262):1562. PMID: 33069326; PMCID: PMC7567026.

Mendonça G, Cheng LA, Mélo EN, de Farias Júnior JC. Physical activity and social support in adolescents: a systematic review. Health Educ Res. 2014 Oct;29(5):822-39. doi: 10.1093/her/cyu017. Epub 2014 May 8. PMID: 24812148.

Cheng LA, Mendonça G, Farias Júnior JC. Physical activity in adolescents: analysis of the social influence of parents and friends. J Pediatr (Rio J). 2014 Jan-Feb;90(1):35-41. doi: 10.1016/j.jped.2013.05.006. Epub 2013 Oct 22. PMID: 24156835.

Su DLY, Tang TCW, Chung JSK, Lee ASY, Capio CM, Chan DKC. Parental Influence on Child and Adolescent Physical Activity Level: A Meta-Analysis. Int J Environ Res Public Health. 2022 Dec 15;19(24):16861. doi: 10.3390/ijerph192416861. PMID: 36554746; PMCID: PMC9778652.

Camargo EM de, Santos MPM, Ribeiro AGP, Mota J, Campos W de. Interação dos fatores sociodemográficos na associação entre fatores psicossociais e transporte ativo para a escola. Cad Saúde Pública [Internet]. 2020;36(Cad. Saúde Pública, 2020 36(5)):e00102719. Available from: https://doi.org/10.1590/0102-311X00102719

  1. The Methods section is well-written and provides all the necessary details for the study. However, the authors could benefit from providing more information on the instruments used to measure physical activity and social support. They could also discuss the validity and reliability of these instruments.

Answer = Thank you for your indication. The following information has been added:

Page 3= “Social support (SS) from parents and friends for physical activity was measured by a 10-item scale (ASAFA Scale), which shows satisfactory internal consistency (parents: α ≥ 0.77 and CFI ≥ 0.83; friends: α ≥ 0.87 and CFI ≥ 0.91) [25]”.

Page 4: “The level of physical activity of adolescents was measured by the Questionnaire of Physical Activity for Adolescents (QAFA) [26], which was first developed in a checklist format by Sallis et al. [27] for US adolescents, with later translation and adaptation for Brazilian adolescents by Farias-Júnior and collaborators [26]. The questionnaire is composed of a list of 24 moderate to vigorous physical activities, with the possibility of the adolescent adding activities beyond those listed. In Brazilian adolescents, the questionnaire showed good reproducibility (intraclass correlation coefficient = 0.88; 95% CI = 0.84-0.91) and concurrent validity in comparison with 24-hour recall (r = 0.62; p < 0.001) [26].

  1. The authors could provide a more detailed description of the statistical methods used in the analysis. For example, it is not clear how the authors determined which categories showed the greatest difference in proportions between the sexes when using Pearson's chi-square test. A more detailed explanation of the post hoc tests used would help readers understand the results better.
  2. The authors could provide more information on the confirmatory factor analysis (CFA) and structural equation modeling (SEM) procedures. For example, it is not clear how the authors determined the number of factors to extract or the justification for using a particular SEM. A more detailed description of the CFA and SEM procedures would help readers understand the validity of the model. The authors could provide more information on the Weighted Least Square means and variance adjusted (WLSMV) estimator used in the analysis. For example, it is not clear how this estimator handles categorical variables or why it was chosen for this analysis.

Answer = Thank you for your indication. Descriptive statistics were performed to verify the sample characteristics and social support (SPSS software version 23).

The following information has been added:

Page 4: “Due to the use of dichotomous variables, the Weighted Least Square means and variance adjusted (WLSMV) estimator was used, which provides the calculation for categorical correlations. The model was considered valid when it met the following criteria: a) value of the chi-square (X2) adjustment statistic with p > 0.05; b) root mean error of approximation (RMSEA) ≤ 0.05 [28]. At the literature-recommended measures of Comparative Fit index (CFI) ≥ 0.90 and Tucker‒Lewis index (TLI)) ≥ 0.90, the estimated models did not reach such high values, but acceptable values for chi-square and RMSEA [28]”. The model fit index values of CFI and TLI are commonly used above 0.90. The mean square error of approximation (RMSEA) must be less than 0.08, which indicates an acceptable fit of the model. Values less than 0.05 are considered very good. The comparative fit index (CFI) must be greater than 0.90, which indicates a good model fit (Kline, 2016).

28-Kline, R. B. (2016). Principles and practice of structural equation modeling (4th ed.) New York, NY The Guilford Press.

  1. The authors could provide more details on the complex sampling plan used to correct for clustering effects. For example, it is not clear how the authors determined the codes used for administrative regions and schools.

Answer = Thank you for your comment.

Page 2 = For the sample calculation regarding the association, a RP=1.49 between social support and the practice of physical activities, a confidence level of 95% (α = 0.05) with a power of 80% (β = 0.20). To correct the error related to the sample selection process, an inflation factor (i.e., design effect) of 1.4 was considered to account for variation in cluster size. Furthermore, an addition of 30% for possible losses and refusals was considered, which estimated a minimum sample required for the study of 965 adolescents. These calculations were performed in the G*Power calculator version 3.1.9.4 (G*Power, Dusseldorf, Germany).

The sample was selected from the multistage, three-stage sampling process: 1st stage - all state schools were stratified according to each of the nine administrative regions of the municipality of Curitiba; 2nd stage - a draw of two schools in each of the nine administrative regions of the city was carried out; 3rd stage: a simple random selection of one class of each year of high school was carried out, according to the number of schoolchildren, separated by sex, required for a given administrative region of the municipality. Finally, all students in each class were invited to participate in the study.

Whitley E, Ball J. Statistics review 4: Sample size calculations. Crit Care. 2002;6(4):335–41.

  1. The authors could provide more detail about the specific findings of their study, rather than simply stating that the results are in line with those published in the literature. For example, they could describe the magnitude of the effects observed and any notable patterns in the data.

Answer = Thank you for your indication. The following information has been added:

Page 8= “Receiving support from parents and friends for physical activity practice, regardless of the amount of physical activity performed by adolescents in a usual week, increases the chances of practicing physical activity. The probability did not increase linearly to any of the sources (parents and friends).

Support from parents has been related to, more than the support received from friends, the probability of fulfilling the amount of adolescents’ physical activity, regardless of the amount performed 180, 300 or 420 min/week, which strengthens the results found in the literature [7,8,17,18]. Otherwise, studies have shown that during childhood, children more frequently perceive the social support received from parents, and in adolescence, they more frequently perceive the support received from friends [16]. It is noteworthy that, regardless of adolescence being a period of greater family independence, their social support seems to play an important role in increasing the level of physical activity, which highlights the need for family support, especially regarding transportation, positive feedback for physical activity and logistic support for transport [7,8,16-19,31,32]

Page 9= Although adolescence experiences vary according to context [19], the family promotes changes in behavioral patterns, and adolescents who perceive their support for physical activity practice end up adopting healthy habits and becoming physically active [32-36]. The family, especially parents, teaches skills and holds beliefs that help adolescents shape important health attitudes and behaviors, most specifically with physical activity and reducing sedentary behavior [32-36]. In a systematic review of reviews [36], in which the included studies were rated with very high quality and relevance, evidence revealed the key role played by parents in promoting physical activity during childhood and adolescence.

  1. The authors could more explicitly address the limitations of their study and the implications of these limitations for the interpretation of their results. For example, they could discuss how the cross-sectional design of the study limits the ability to draw causal conclusions and the potential impact of selection bias due to the use of a convenience sample.

Answer = Thank you for your indication. The following information has been added:

Page 09= “Some limitations should be considered for a better understanding of the results. Reverse causality, a common characteristic in studies with a cross-sectional design, does not allow establishing a cause-and-effect relationship or the direction of relationships. However, the need for longitudinal studies examining this association is important. Moreover, the study was developed in a single Brazilian city that presents typical characteristics of well-developed urban centers, which does not allow extrapolation of the findings to rural centers and other cities in the country. In addition, the sample consisted only of adolescent students from the public school system, which makes it impossible to extrapolate the results to higher classes. However, the representative sample and statistical analyses ensure an interpretation of the data for large public school populations. Additionally, the use of self-reported measures could be influenced by social desirability or recall bias. To minimize this bias, researchers were trained to assist adolescents in answering the questionnaires. The instrument used to measure physical activity does not allow us to identify all the domains and contexts in which the activities were practiced; for this reason, we were able to obtain a more objective measurement of the physical activity level. However, because this was a large study with a representative sample, the first study in assessing this association, the use of questionnaires proved to be the best alternative”.

  1. The authors could provide more context for their findings by discussing how they relate to previous research in the field. For example, they could compare their results to those of other studies that have examined the role of social support in promoting physical activity among adolescents, and they could discuss any inconsistencies or discrepancies between their findings and those of previous research.

Answer = Thank you for your indication. The following information has been added:

Page 8= “Support from parents has been related to, more than the support received from friends, the probability of adolescents engaging in physical activity, regardless of the amount performed 180, 300 or 420 min/week, which strengthens the results found in the literature [7,8,17,18]. Otherwise, studies have shown that during childhood, children more frequently perceive the social support received from parents, and in adolescence, they more frequently perceive the support received from friends [16]. It is noteworthy that, regardless of adolescence being a period of greater family independence, their social support seems to play an important role in increasing the level of physical activity, which highlights the need for family support, especially regarding transportation, positive feedback for physical activity and logistic support for transport [7,8,16-19,31,32]. The studies believe that this family support may vary according to the region of the country [31,32], which may partially explain the difference found in the data obtained among adolescents from Curitiba (Paraná, Brazil)”.

  1. The authors could more explicitly state the practical implications of their findings for public health policy and intervention development. For example, they could discuss how their findings could inform the design of programs aimed at promoting physical activity among adolescents, and they could suggest specific strategies that could be employed to increase social support for physical activity among parents and friends.

Answer = Thank you for your indication. The following information has been added:

Page 9= As a practical application, our results seem to highlight the need: parents should encourage adolescents to be physically active in their leisure time [7,8,16]. Suggesting changes in the adolescent's daily routine aimed at maximizing energy expenditure may be a good alternative to help adolescents realize opportunities to be active during their daily routine, for example, going for a walk at the end of the day on a skateboard, bicycle, on foot; exercising in public leisure spaces; or even visiting public spaces through outings with family and friends [7,16,19,31,32]. The incentive for the practice of physical activity in leisure time, even with small insertions in the daily routine, may contribute to increasing the amount of physical activity in a usual week [7,8,16].

Page 9= “As a practical application, our results seem to highlight the need: group discussion, conversations, e-mail messages and apps, and reading texts about the behavior can be efficient strategies to disseminate the practice. Verbal persuasion activities that prompt adolescents to express their beliefs in their ability to adopt the behavior may support a better understanding of actual outcome expectations. Another suggested strategy is to visualize people or groups exercising in public places, which can increase the motivation to perform the activity [9,10]. These techniques can be introduced in interventions that aim to increase physical activity practice in different contexts, such as physical education classes at school, LTPA, active commuting to school, and active leisure time at school [19,37]. School is the place where adolescents spend most of their time away from home and in the company of their friends [37]. Their social support plays an important role in the relationship with physical activity since friends share ideas, habits, and beliefs, which can increase the amount of physical activity when friends are also physically active [19,37].”

  1.  Overall, the manuscript appears to be well-written and presents important findings on the association between social support and physical activity among adolescents. The study design, including a representative sample of 1984 adolescents and the use of validated scales for social support and physical activity, adds credibility to the results. One potential improvement for the manuscript would be to provide more detailed information on the conceptual model for structured equations that was used in the statistical analysis. This would help readers better understand how the authors arrived at their conclusions.

Additionally, while the study found that social support from parents and friends increased the likelihood of adolescents engaging in different amounts of leisure-time physical activity, the manuscript could benefit from a discussion on the implications of these findings. For example, what strategies could parents and friends use to provide incentives and support for physical activity? How can schools and public health professionals promote social support for physical activity among adolescents?

Overall, this manuscript is a valuable contribution to the field of adolescent physical activity and social support, and the authors should be commended for their work.

Answer = Thank you for your indication. The following information has been added:

Page 10= “The results identified are important to enhance future interventions with the purpose of changing behavior in the amount of physical activity among adolescents. As a public health message, highlight as implications of the research a set of actions aimed at disseminating and maximizing the practice of physical activity among adolescents. Good strategies to encourage practice include education by example aimed at incorporating attitudes and behaviors that facilitate physical activity and reducing sedentary behavior; guidance to perceive opportunities to stay physically active in their neighborhood through knowledge of new public places with their friends; and participation in social groups that involve discussion of the topic through conversations, messages, informative texts, and the use of social media for interaction with people of the same age on topics relevant to physical activity, exercise, and sports.”

Thank you for your comment. As previously mentioned, the conceptual model for SEM analysis has been further explained in the statistical section.

As previously explained in question 11, strategies parents and friends use to provide incentives and support for physical activity were rewritten.

References

  1. Guthold, R., Stevens, G. A., Riley, L. M., & Bull, F. C. (2020). Global trends in insufficient physical activity among adolescents: a pooled analysis of 298 population-based surveys with 1·6 million participants. The Lancet. Child & adolescent health4(1), 23–35. https://doi.org/10.1016/S2352-4642(19)30323-2
  2. GBD 2019 Mental Disorders Collaborators (2022). Global, regional, and national burden of 12 mental disorders in 204 countries and territories, 1990-2019: a systematic analysis for the Global Burden of Disease Study 2019. The lancet. Psychiatry9(2), 137–150. https://doi.org/10.1016/S2215-0366(21)00395-3
  3. GBD 2019 Diseases and Injuries Collaborators (2020). Global burden of 369 diseases and injuries in 204 countries and territories, 1990-2019: a systematic analysis for the Global Burden of Disease Study 2019. Lancet (London, England)396(10258), 1204–1222. https://doi.org/10.1016/S0140-6736(20)30925-9
  4. NCD Risk Factor Collaboration (NCD-RisC) (2017). Worldwide trends in body-mass index, underweight, overweight, and obesity from 1975 to 2016: a pooled analysis of 2416 population-based measurement studies in 128·9 million children, adolescents, and adults. Lancet (London, England)390(10113), 2627–2642. https://doi.org/10.1016/S0140-6736(17)32129-3
  5. Aubert, S., Barnes, J. D., Abdeta, C., Abi Nader, P., Adeniyi, A. F., Aguilar-Farias, N., Andrade Tenesaca, D. S., Bhawra, J., Brazo-Sayavera, J., Cardon, G., Chang, C. K., Delisle Nyström, C., Demetriou, Y., Draper, C. E., Edwards, L., Emeljanovas, A., Gába, A., Galaviz, K. I., González, S. A., Herrera-Cuenca, M., … Tremblay, M. S. (2018). Global Matrix 3.0 Physical Activity Report Card Grades for Children and Youth: Results and Analysis From 49 Countries. Journal of physical activity & health15(S2), S251–S273. https://doi.org/10.1123/jpah.2018-0472
  6. Garcia-Hermoso, A., López-Gil, J. F., Ramírez-Vélez, R., Alonso-Martínez, A. M., Izquierdo, M., & Ezzatvar, Y. (2023). Adherence to aerobic and muscle-strengthening activities guidelines: a systematic review and meta-analysis of 3.3 million participants across 32 countries. British journal of sports medicine57(4), 225–229. https://doi.org/10.1136/bjsports-2022-106189
  7. Laird, Y., Fawkner, S., Kelly, P., McNamee, L., & Niven, A. (2016). The role of social support on physical activity behaviour in adolescent girls: a systematic review and meta-analysis. The international journal of behavioral nutrition and physical activity13, 79. https://doi.org/10.1186/s12966-016-0405-7
  8. Laird, Y., Fawkner, S., & Niven, A. (2018). A grounded theory of how social support influences physical activity in adolescent girls. International journal of qualitative studies on health and well-being13(1), 1435099. https://doi.org/10.1080/17482631.2018.1435099
  9. Tcymbal, A., Demetriou, Y., Kelso, A., Wolbring, L., Wunsch, K., Wäsche, H., Woll, A., & Reimers, A. K. (2020). Effects of the built environment on physical activity: a systematic review of longitudinal studies taking sex/gender into account. Environmental health and preventive medicine25(1), 75. https://doi.org/10.1186/s12199-020-00915-z
  10. Nordbø, E. C. A., Nordh, H., Raanaas, R. K., & Aamodt, G. (2020). Promoting activity participation and well-being among children and adolescents: a systematic review of neighborhood built-environment determinants. JBI evidence synthesis18(3), 370–458. https://doi.org/10.11124/JBISRIR-D-19-00051
  11. Tassitano, R. M., Weaver, R. G., Tenório, M. C. M., Brazendale, K., & Beets, M. W. (2020). Physical activity and sedentary time of youth in structured settings: a systematic review and meta-analysis. The international journal of behavioral nutrition and physical activity17(1), 160. https://doi.org/10.1186/s12966-020-01054-y
  12. Rosselli, M., Ermini, E., Tosi, B., Boddi, M., Stefani, L., Toncelli, L., & Modesti, P. A. (2020). Gender differences in barriers to physical activity among adolescents. Nutrition, metabolism, and cardiovascular diseases : NMCD30(9), 1582–1589. https://doi.org/10.1016/j.numecd.2020.05.005
  13. Martínez-Andrés, M., Bartolomé-Gutiérrez, R., Rodríguez-Martín, B., Pardo-Guijarro, M. J., Garrido-Miguel, M., & Martínez-Vizcaíno, V. (2020). Barriers and Facilitators to Leisure Physical Activity in Children: A Qualitative Approach Using the Socio-Ecological Model. International journal of environmental research and public health, 17(9), 3033. https://doi.org/10.3390/ijerph17093033
  14. Sallis, J. F., Prochaska, J. J., & Taylor, W. C. (2000). A review of correlates of physical activity of children and adolescents. Medicine and science in sports and exercise, 32(5), 963–975. https://doi.org/10.1097/00005768-200005000-00014
  15. Sallis, J.F.; Owen, N.; Fisher, E.B. Ecological models of health behavior. In: Glanz K, Rimer BK, Viswanath K, editors. Health behavior and health education: theory, research, and practice. 4th ed. San Francisco, CA: Jossey-Bass, 2008, 465-485.
  16. Mendonça, G., Cheng, L. A., Mélo, E. N., & de Farias Júnior, J. C. (2014). Physical activity and social support in adolescents: a systematic review. Health education research29(5), 822–839. https://doi.org/10.1093/her/cyu017
  17. Cheng, L. A., Mendonça, G., & Farias Júnior, J. C. (2014). Physical activity in adolescents: analysis of the social influence of parents and friends. Jornal de pediatria90(1), 35–41. https://doi.org/10.1016/j.jped.2013.05.006
  18. Su, D. L. Y., Tang, T. C. W., Chung, J. S. K., Lee, A. S. Y., Capio, C. M., & Chan, D. K. C. (2022). Parental Influence on Child and Adolescent Physical Activity Level: A Meta-Analysis. International journal of environmental research and public health19(24), 16861. https://doi.org/10.3390/ijerph192416861
  19. Camargo, E. M. de ., Santos, M. P. M., Ribeiro, A. G. P., Mota, J., & Campos, W. de .. (2020). Interação dos fatores sociodemográficos na associação entre fatores psicossociais e transporte ativo para a escola. Cadernos De Saúde Pública, 36(Cad. Saúde Pública, 2020 36(5)), e00102719. https://doi.org/10.1590/0102-311X00102719
  20. Bull, F. C., Al-Ansari, S. S., Biddle, S., Borodulin, K., Buman, M. P., Cardon, G., Carty, C., Chaput, J. P., Chastin, S., Chou, R., Dempsey, P. C., DiPietro, L., Ekelund, U., Firth, J., Friedenreich, C. M., Garcia, L., Gichu, M., Jago, R., Katzmarzyk, P. T., Lambert, E., … Willumsen, J. F. (2020). World Health Organization 2020 guidelines on physical activity and sedentary behaviour. British journal of sports medicine54(24), 1451–1462. https://doi.org/10.1136/bjsports-2020-102955
  21. Chaput, J. P., Willumsen, J., Bull, F., Chou, R., Ekelund, U., Firth, J., Jago, R., Ortega, F. B., & Katzmarzyk, P. T. (2020). 2020 WHO guidelines on physical activity and sedentary behaviour for children and adolescents aged 5-17 years: summary of the evidence. The international journal of behavioral nutrition and physical activity17(1), 141. https://doi.org/10.1186/s12966-020-01037-z
  22. de Onis, M., Onyango, A. W., Borghi, E., Siyam, A., Nishida, C., & Siekmann, J. (2007). Development of a WHO growth reference for school-aged children and adolescents. Bulletin of the World Health Organization85(9), 660–667. https://doi.org/10.2471/blt.07.043497
  23. Tanner, J.M. Growth at adolescence. 2. ed. Oxford: Blackwell, 1962.
  24. Associação Brasileira de Empresas de Pesquisa. Critério de classificação econômica Brasil. São Paulo, 2016, 1-6.
  25. Farias Júnior, J.C.; Mendonça, G.; Florindo, A.A.; Barros, M.V.G. Fidedignidade e validade de uma escala de avaliação do apoio social para prática de atividade física para adolescentes: Escala ASAFA. Bras. Epidemiol 2014, 17, 355-70. https://doi.org/10.1590/1809-4503201400020006.
  26. Farias Júnior, J.C.; Lopes, A.S.; Mota, J.; Santos, M.P.; Ribeiro, J.C.; Hallal, P.C. Validade e reprodutibilidade de um questionário para medida de atividade física em adolescentes: uma adaptação do Self-Administered Physical Activity Checklist. Bras. Epidemiol. 2012, 15, 198-210. https://doi.org/10.1590/S1415-790X2012000100018.
  27. Sallis, J. F., Strikmiller, P. K., Harsha, D. W., Feldman, H. A., Ehlinger, S., Stone, E. J., Williston, J., & Woods, S. (1996). Validation of interviewer- and self-administered physical activity checklists for fifth grade students. Medicine and science in sports and exercise28(7), 840–851. https://doi.org/10.1097/00005768-199607000-00011
  28. Kline, R. B. (2016). Principles and practice of structural equation modeling (4th ed.) New York, NY The Guilford Press.
  29. Marshall, S. J., & Biddle, S. J. (2001). The transtheoretical model of behavior change: a meta-analysis of applications to physical activity and exercise. Annals of behavioral medicine : a publication of the Society of Behavioral Medicine23(4), 229–246. https://doi.org/10.1207/S15324796ABM2304_2
  30. Spencer, L., Adams, T. B., Malone, S., Roy, L., & Yost, E. (2006). Applying the transtheoretical model to exercise: a systematic and comprehensive review of the literature. Health promotion practice7(4), 428–443. https://doi.org/10.1177/1524839905278900
  31. Lisboa, T., Silva, W. R. D., Silva, D. A. S., Felden, É. P. G., Pelegrini, A., Lopes, J. J. D., & Beltrame, T. S. (2021). Social support from family and friends for physical activity in adolescence: analysis with structural equation modeling. Cadernos de saude publica37(1), e00196819. https://doi.org/10.1590/0102-311X00196819
  32. Khan, A., & Uddin, R. (2020). Parental and peer supports are associated with an active lifestyle of adolescents: evidence from a population-based survey. Public health188, 1–3. https://doi.org/10.1016/j.puhe.2020.08.024
  33. Lawler, M., Heary, C., & Nixon, E. (2020). Peer Support and Role Modelling Predict Physical Activity Change among Adolescents over Twelve Months. Journal of youth and adolescence49(7), 1503–1516. https://doi.org/10.1007/s10964-019-01187-9
  34. Petersen, T. L., Møller, L. B., Brønd, J. C., Jepsen, R., & Grøntved, A. (2020). Association between parent and child physical activity: a systematic review. The international journal of behavioral nutrition and physical activity17(1), 67. https://doi.org/10.1186/s12966-020-00966-z
  35. Pluta, B., Korcz, A., Krzysztoszek, J., Bronikowski, M., & Bronikowska, M. (2020). Associations between adolescents' physical activity behavior and their perceptions of parental, peer and teacher support. Archives of public health = Archives belges de sante publique78, 106. https://doi.org/10.1186/s13690-020-00490-3
  36. Messing, S., Rütten, A., Abu-Omar, K., Ungerer-Röhrich, U., Goodwin, L., Burlacu, I., & Gediga, G. (2019). How Can Physical Activity Be Promoted Among Children and Adolescents? A Systematic Review of Reviews Across Settings. Frontiers in public health7, 55. https://doi.org/10.3389/fpubh.2019.00055
  37. Hoehner, C. M., Ribeiro, I. C., Parra, D. C., Reis, R. S., Azevedo, M. R., Hino, A. A., Soares, J., Hallal, P. C., Simões, E. J., & Brownson, R. C. (2013). Physical activity interventions in Latin America: expanding and classifying the evidence. American journal of preventive medicine44(3), e31–e40. https://doi.org/10.1016/j.amepre.2012.10.026

Round 2

Reviewer 3 Report

Although there are still stylistic corrections in the text, most of the suggested changes have been made.

Author Response

Dear,

Thank you for your comment. 

Kind regards,

Reviewer 4 Report

All corrections are made 

Author Response

(The authors gave the same response as above.)
